# Socio-ecological risk factors associated with human flea infestations of rural household in plague-endemic areas of Madagascar

Adélaïde Miarinjara[1]*, Annick Onimalala Raveloson[2,3], Stephen Gilbert Mugel[1], Nick An[1], Andry Andriamiadanarivo[4], Minoarisoa Esther Rajerison[5], Rindra Vatosoa Randremanana[6], Romain Girod[2], Thomas Robert Gillespie[1,4]*

**1** Departments of Environmental Sciences and Environmental Health, Emory University and Rollins School of Public Health, Atlanta, United States of America, **2** Medical Entomology Unit, Institut Pasteur de Madagascar, Antananarivo, Madagascar, **3** Ecole Doctorale Science de la Vie et de l'Environnement, Université d'Antananarivo, Antananarivo, Madagascar, **4** Centre Valbio, Ranomafana, Madagascar, **5** Plague Unit, Institut Pasteur de Madagascar, Antananarivo, Madagascar, **6** Epidemiological and Clinical Research Unit, Institut Pasteur de Madagascar, Antananarivo, Madagascar

\* adelaide.miarinjara@emory.edu (AM); thomas.gillespie@emory.edu (TRG)

**Data Availability Statement:** All data generated or analyzed during this study are included in this published article and its supplementary information

## Abstract

Plague is a flea-borne fatal disease caused by the bacterium *Yersinia pestis*, which persists in rural Madagascar. Although fleas parasitizing rats are considered the primary vectors of *Y. pestis*, the human flea, *Pulex irritans*, is abundant in human habitations in Madagascar, and has been found naturally infected by the plague bacterium during outbreaks. While *P. irritans* may therefore play a role in plague transmission if present in plague endemic areas, the factors associated with infestation and human exposure within such regions are little explored. To determine the socio-ecological risk factors associated with *P. irritans* infestation in rural households in plague-endemic areas of Madagascar, we used a mixed-methods approach, integrating results from *P. irritans* sampling, a household survey instrument, and an observational checklist. Using previously published vectorial capacity data, the minimal *P. irritans* index required for interhuman bubonic plague transmission was modeled to determine whether household infestations were enough to pose a plague transmission risk. Socio-ecological risk factors associated with a high *P. irritans* index were then identified for enrolled households using generalized linear models. Household flea abundance was also modeled using the same set of predictors. A high *P. irritans* index occurred in approximately one third of households and was primarily associated with having a traditional dirt floor covered with a plant fiber mat. Interventions targeting home improvement and livestock housing management may alleviate flea abundance and plague risk in rural villages experiencing high *P. irritans* infestation. As plague-control resources are limited in developing countries such as Madagascar, identifying the household parameters and human behaviors favoring flea abundance, such as those identified in this study, are key to developing preventive measures that can be implemented at the community level.

files, as well as at OSF, under a public repository link (https://osf.io/6mqux/?view_only=70d21dff44ce42bc8fc9bc45d8a9bf8c) in the manuscript main text.

**Funding:** AM received funding from The Branco Weiss Fellowship Society in Science, (https://brancoweissfellowship.org) and The American Society of Tropical Medicine and Hygiene, through the Centennial Travel Award (https://www.astmh.org/awards-fellowships-medals/awards-and-honors/centennial-travel-award). The funders had no role in study design, data collection and analysis, decision to publish, or preparation of the manuscript.

**Competing interests:** The authors have declared that no competing interests exist.

## Author summary

Plague is a bacterial disease transmitted by flea bites, and the rat fleas are the main vectors of *Yersinia pestis*, the plague bacterium. Households in plague endemic-areas of Madagascar are frequently infested by *Pulex irritans*, the human flea, which does become naturally infected with the plague bacterium during epidemic. The intensity of flea infestation varies among households, but the reasons for such disparities are poorly understood. This study identifies factors associated with *P. irritans* infestation in rural households in plague-endemic areas of Madagascar. Infestation risk was more pronounced for poor households living in homes made with organic construction materials and flea density did not show a seasonal pattern. One third of the household experienced high flea infestation, putting inhabitants at risk of sustained interhuman plague transmission, should the fleas or a household member become infected. While *P. irritans* may be a secondary vector, this additional route of plague transmission deserves more attention from epidemiologists. The factors identified in this analysis suggest that improvement of housing and better management of livestock would alleviate flea burden and potential plague risk in rural plague-endemic villages experiencing high flea infestation.

## Background

Fleas (Order Siphonaptera) are bloodsucking, wingless insects with laterally- compressed bodies and hind legs specialized for jumping [1]. Flea species from Pulicidae and Tungidae families are important pests for humans and domestic animals and include species such as *Xenopsylla cheopis* (the Oriental rat flea), *Pulex irritans* (the human flea), *Ctenocephalides canis* and *Ctenocephalides felis* (dog and cat fleas, respectively), *Echidnophaga gallinacea* (the stick-tight flea), and *Tunga penetrans* (jigger flea), which are commonly found in the human environment [2]. In many cases, flea infestations are concurrent among livestock and companion animals, which act as reservoir and/or principal hosts [3–5]. Flea infestations have not received much attention despite their detrimental impacts on community morbidity, wellbeing, and productivity in low-income countries [6–8].

Fleas undergo full metamorphosis and the immature flea life-stages live among the dust and crevices of floors within homes or in animal host burrows. The photophobic worm-like larvae require high humidity to survive and feed on various organic debris in the environment. Flea life cycle from egg to adult is influenced by factors related to the immediate environment, such as temperature, humidity, and host presence [2,9,10]. Adult fleas of both sexes feed exclusively on blood. Some flea species live in animal nests and burrows as adults, while others live on host fur, leaving only if the host dies. Host blood source has a decisive impact on flea population maintenance since blood components determine flea fitness and survival [11]. Flea host specificity (i.e., number of host species exploited) depends on factors that affect both adult and immature stage survival; and host availability and preferences determine flea distributions and their role in the transmission of pathogenic parasites and bacteria [9]. Cat and dog fleas serve as intermediate hosts for various tapeworms (Order Cyclophyllidea), contributing to the spread of the parasite among companion animals and potential zoonotic exposure [9]. Murine typhus is a flea-borne rickettsial disease caused by infection with *Rickettsia typhi* that has been detected in flea species including *C. felis*, *E. gallinacea*, *P. irritans*, and *X. cheopis* [12,13]. Fleas may also play a limited role in the transmission of tularemia, a bacterial disease caused by *Francisella tularensis* [9]. *Bartonella sp*., responsible for bartonellosis, has been detected in various flea species parasitizing commensal and wild hosts [2].

Among flea-borne diseases, plague is arguably the most infamous [2]. *Yersinia pestis*, the etiologic agent of plague, is a highly virulent bacterium that has killed millions during three historic human pandemics and continues to re-emerge [14]. The transmission cycle of *Y. pestis* is complex, involving multiple vertebrate hosts. Plague is principally a flea-borne rodent disease, characterized by circulation within resistant rodent populations, inducing low or no mortality but allowing persistence of the pathogen in the environment (enzootic plague), and transmission between susceptible rodent populations inducing high mortality (epizootic) [15]. Humans are most likely to become infected when flea numbers are high and epizootic plague is decimating a susceptible rodent population, as infected fleas from dead rodents are seeking new hosts [16].

In countries such as Uganda, Madagascar, and Tanzania, where bubonic plague is prevalent, *C. felis* and *P. irritans* are among the most abundant fleas in homes and are categorized as house-dwelling, free, host-seeking or house fleas, as opposed to on-host fleas [17–19]. Interestingly, *C. felis*, and the human flea, *P. irritans*, are considered of low concern for public health despite a presumable role in plague transmission [17,20–22]. In Madagascar, rat fleas are the only target of vector control efforts, and solely within the framework of plague epidemic mitigation [19,23,24]. Troublingly the insecticide powder used for flea control during plague outbreaks, spread on the household floor or contained in bait stations, has little effect on *P. irritans* [23,24]. Furthermore, households in plague-endemic areas of Madagascar were frequently infested by a large number of human fleas at magnitudes rarely found in other countries reporting human plague outbreaks [18,25]. Although the intensity of flea infestation varies greatly among households, the reasons for such disparities are poorly understood [19,23]. Research regarding *P. irritans* biology, ecology, and the conditions under which this species may play a role in plague transmission are scarce in Madagascar, though this knowledge would be valuable to develop science-based plague control strategies.

The aim of the present study was to determine the socio-ecological risk factors associated with *P. irritans* infestations that may increase plague transmission risk in rural Madagascar households where plague is known to circulate, or recent outbreaks have occurred. Our primary hypothesis was that *P. irritans* density is driven by seasonal patterns and influenced by household characteristics. As ectoparasite control resources are limited in developing countries, identifying household parameters and human behaviors favoring flea abundance and plague risk are key to developing preventive measures that can be implemented by community engagement. Our specific aims were: 1. to identify household-level characteristics that correspond to high-risk *P. irritans* abundance and 2. to assess seasonal variation in *P. irritans* abundance in homes in plague-endemic region of Madagascar.

## Methods

### Ethic statement

Participants in this study were adults (> 18 years old) that provided oral informed consent for interview and flea sampling in their homes. The project was reviewed and approved by the Emory University Institutional Review Board (STUDY00004288), the Institut Pasteur de Madagascar scientific committee, and the biomedical research ethics committee of the Malagasy Ministry of Public Health (Comité d'Ethique de la Recherche Biomédicale, case number 82-MSANP/SG/AMM/CERBM).

### Study area

Repeated cross-sectional surveys and household flea sampling were conducted in four rural villages within the plague-endemic Southeastern part of the Central Highlands of Madagascar

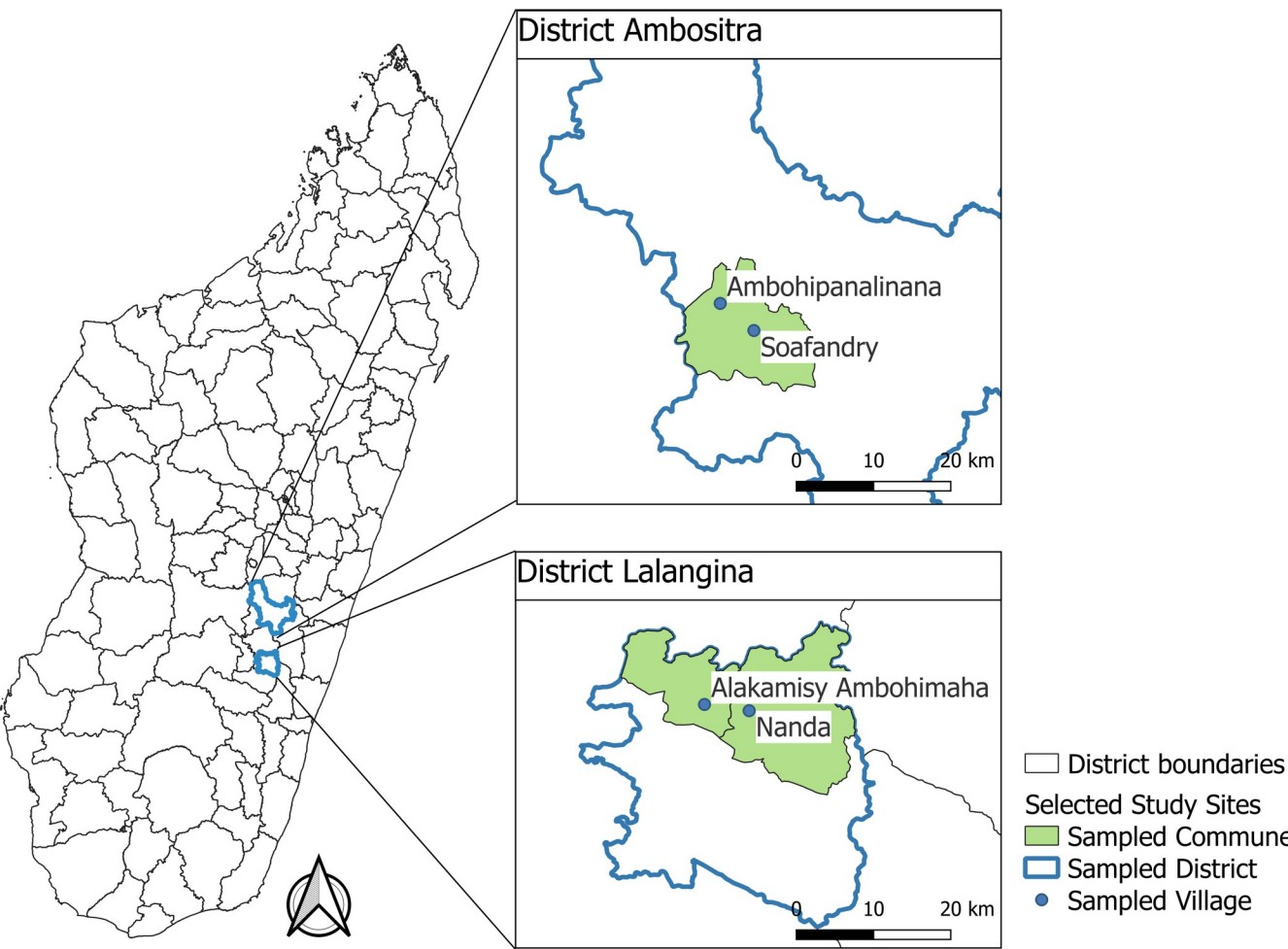

**Fig 1. Map of study sites for investigation of socio-ecological risk factors for rural household flea infestations in plague-endemic areas of Madagascar.**
The map was generated with QGIS software (https://qgis.org/en/site/about/index.html). Administrative boundaries were downloaded from GADM: https://gadm.org/index.html.

[19,26] (Fig 1). Two villages, Nanda and Alakamisy Ambohimaha, belong to the Lalangina district, within the Matsiatra Ambony Region where multiple suspected or confirmed plague outbreaks have been reported in the past decade. The other two villages, Soafandry and Ambohipanalinana, belong to the district of Ambositra (Amoron'i Mania region), which has several active plague foci [27,28].

## Survey instrument

Interviews were conducted during the dry season (June—July 2022) when workload in rice paddies was lowest and heads of households were expected to be available for interviews. The survey instrument was developed in English, translated to Malagasy, and validated via back-translation [29] before being administered orally to each head of household in Malagasy. Village leaders and investigators called open community gatherings where attendees were informed of the purpose, scope, methods, and plans for information sharing of the study, and an investigator disclosed that participation in the study was voluntary and prospective participants were asked for their informed consent. Households were selected randomly, starting from the village gathering

place to the periphery and included when an adult (>18 years old) was present and orally consented to participate to the study. The survey instrument focused on socio-ecological variables that may influence abundance of *P. irritans* in households including demographics, sleeping arrangement, presence of animals, behavioral practices related to home hygiene, and attitudes towards rodents and fleas (S1 File). Observational data related to household characteristics such as building materials and presence/absence of animal housing were also collected.

## Flea sampling

Fleas were sampled twice in the four villages, once during the dry season (June–July 2022) and once during the rainy season (November 2021 in Nanda and October 2022 for the three other villages). Fleas were sampled via candle trap method [30]. Briefly, a candle (21.5 cm in height and 1.5 cm in diameter) was lit in the middle of a pale-colored enamel plate (diameter = 22.5 cm) containing water mixed with a pinch of laundry powder. Fleas attracted by the candlelight fell into soapy water and died. Each household received one candle trap per night for three consecutive nights. Candle traps were placed in a room chosen by each head of the household (typically the bedroom) and lit before bedtime, burning until the wick reached the water level (about eight hours). Fleas were collected the following morning one by one, placed on blotting paper to remove excess water, and stored in separate 1.5 ml vials containing 70% ethanol using fine-tip entomological forceps. Flea species were later identified based on morphology using an identification key [31] and individuals of each species were counted at 25X magnification.

## Data analysis

The factors associated with *P. irritans* abundance in households were analyzed using two approaches where the outcome variable was characterized as raw flea count per household (Model 1) or a binomial categorization of *P. irritans* index per household based on simulated plague transmission risk (Model 2). Variables were selected *a priori* according to relevant scientific literature [3–8] and the research team members' own experiences (see Table 1). Variables were excluded if there was too much homogeneity (<20% in a level) in the dataset and were included in the model if there was low collinearity (assessed using "VIF", variance inflation factor function from the R package "car" [32]). Both Models 1 and 2 were conducted on flea data from the dry season only (when interviews were conducted).

Model 1 focused on risk factors for flea abundance as measured by *P. irritans* count per household (totaled across three consecutive sampling nights) using a generalized linear mixed model, which included a random effect for village and utilized a negative binomial distribution to account for overdispersion in flea counts.

Model 2 explored the potential for flea abundance sufficient for plague transmission [17,33,34] by modeling the risk factors associated with a household exhibiting a density of *P. irritans* per person greater than or equal to that estimated to sustain person-to-person transmission of *Y. pestis* as has been done previously [17], based on *P. irritans* vector competence during early phase transmission [21]. Model assumptions accounting vector competence included that: infectious fleas could locate human hosts, early phase transmission was the primary mode of vector transmission, all hosts were equally sought and bitten by fleas [17,33] and parameters were similar for fleas from different populations. The average number of fleas per person required to sustain transmission ($m$) was modeled as follows:

$$m = R_0 \cdot \left( \frac{r}{(a \cdot b \cdot p^n)} \right) \qquad \text{Eq1}$$

**Table 1. Characteristics of rural households in plague-endemic areas of Madagascar.**

| Variables (total number of observations) | Factors | Observation | Proportion (%) |
|---|---|---|---|
| Respondent gender (n = 95) | Female | 79 | 83.16 |
| | Male | 16 | 14.81 |
| Head of household gender (n = 104) | Female | 19 | 18.27 |
| | Male | 85 | 81.73 |
| Head of household marital status (n = 104) | Married | 79 | 75.96 |
| | Single Female | 19 | 18.27 |
| | Single Male | 6 | 5.77 |
| Head of household finished primary school (n = 97) * | Yes | 40 | 41.24 |
| | No | 57 | 58.76 |
| For couples, spouse finished primary school (n = 79) | Yes | 36 | 45.57 |
| | No | 43 | 54.43 |
| Household size (n = 104) * | 1 to 4 | 53 | 50.96 |
| | >4 | 51 | 49.04 |
| House cleaned daily (n = 104) | Yes | 87 | 83.65 |
| | No | 17 | 16.35 |
| Head of household reported rodent activities in house within last two months (n = 104) * | Yes | 79 | 75.96 |
| | No | 25 | 24.04 |
| Household used insecticide to control pests (n = 104) * | Yes | 84 | 80.77 |
| | No | 20 | 19.23 |
| At least one household member sleeps on floor (n = 104) * | Yes | 43 | 41.35 |
| | No | 61 | 58.65 |
| At least one household member sleeps under a bed net (n = 104) * | Yes | 72 | 69.23 |
| | No | 32 | 30.77 |
| Roof type (n = 104) * | Thatch | 35 | 33.65 |
| | Clay tiles | 36 | 34.62 |
| | Metal sheets | 33 | 31.73 |
| Floor type (n = 104) * | Mat | 69 | 66.35 |
| | Board | 24 | 23.08 |
| | Concrete | 6 | 5.77 |
| | Other | 5 | 4.81 |
| Livestock kept near house (n = 103) | Yes | 59 | 57.28 |
| | No | 44 | 42.72 |
| Owning livestock (n = 104) | Yes | 89 | 85.58 |
| | No | 15 | 14.42 |
| Livestock kept indoors at night (n = 104) | Yes | 82 | 78.85 |
| | No | 22 | 21.15 |
| Chickens kept indoors at night (n = 104) * | Yes | 65 | 62.50 |
| | No | 39 | 37.50 |
| Other poultry kept indoors at night (n = 104) | Yes | 27 | 25.96 |
| | No | 77 | 74.04 |
| Pigs kept indoors at night (n = 104) * | Yes | 33 | 31.73 |
| | No | 71 | 68.27 |
| Cows kept indoors at night (n = 104) * | Yes | 35 | 33.65 |
| | No | 69 | 66.35 |
| Rabbits and guinea pigs kept indoors at night (n = 104) | Yes | 21 | 20.19 |
| | No | 83 | 79.81 |

* Variables included in both models.

In the equation, $R_0$ represented the average number of secondary infections and was set at $R_0 = 1$ to model minimum sustained transmission at a population level [34]. The daily biting rate of *P. irritans*, *a*, was described using a beta distribution based on a recent laboratory study in which 230 of 280 *P. irritans* fed daily on human blood [21]. The probability of acquiring and transmitting *Y. pestis* during early phase transmission (within 24-hours of infectious blood meal), *b*, was described using a beta distribution based on the same study, which found that 15 of 181 *P. irritans* transmitted *Y. pestis*. The probability of *P. irritans* surviving the extrinsic incubation period (here, the 24-hours of early phase transmission), $p^n$, was estimated to be one, because nearly all fleas survived this short period [17,21]. The average life expectancy of the human host following the threshold septicemia, $1/r$, was estimated as two days based on previous reports [27,35]. Then, using 10,000 simulated random draws from beta distributions for *a* and *b*, a distribution for *m* was generated using R Studio software [32].

Households were categorized as at "higher infestation risk" when $P_{ii}$ (*P. irritans* index, the total number of *P. irritans* collected in a household across three nights divided by the household size) $\geq$ mean (*m*) or households were categorized at "lower infestation risk" when $P_{ii} <$ mean (*m*), referring to household plague transmission via *P. irritans* if a single person were infected.

A binomial generalized linear model was used to identify risk factors for a household being at "higher infestation risk" based on this $P_{ii}$ categorization from the dry season. Village was treated as a fixed effect because the model failed to converge with random effects. A sensitivity analysis was performed with lower and upper 95% confidence intervals of *m*. Seasonality of *P. irritans* was assessed by comparing total flea number per village between seasons using an ANOVA test. All analyses were conducted using R Studio software [32]. R codes and data are available at OSF.

## Results

A total of 126 households were visited in the four villages. Of the households visited, 82.54% participated in all components of the study during both wet and dry seasons. Five heads of household declined to participate in the interview but participated in the flea sampling component. In addition, 12 households only participated in one season of flea sampling, and an additional five households ended involvement prematurely. Only households that gave interview consent and where flea sampling was conducted during both seasons were included in analyses (n = 104).

Mean household size was four and ranged from one to 14. Most respondents were women (83.16%), but men (i.e., husband, father, son, or older brother) were usually identified as head of household (81.73%). Among household heads, 24.04% were identified as single (female = 18.27%, male = 5.77%), 58.76% had not finished primary school, 41.35% reported having at least one household member sleeping on the floor (without an elevated bedframe), and 69.23% reported at least one household member using a mosquito net at night on a regular basis.

Most enrolled households resided in houses constructed in traditional fashion (S2 File) for the southern part of the Central Highlands of Madagascar with three-stories (ground floor, first floor and attic) and at least two rooms per level (S3 File). Residents usually slept on the first floor and the kitchen was usually in the attic. Most families kept livestock on the ground floor at night (78.85%), and animal enclosures and pens were observed in proximity to many homes (57.28%). Most households (85.58%) owned livestock, with an average of nine animals per household. A summary of household characteristics is presented in Table 1.

Most floors were either dirt covered with a woven plant fiber mat (66.35%), wooden boards (23.08%), or concrete (5.77%). Other material types such as vinyl sheets and tarps were

**Table 2. Flea nuisance perception and insecticide use in plague-endemic areas of Madagascar.**

| Variable (total number of observations) | Factor | Observation | Frequency (%) |
|---|---|---|---|
| Flea nuisance intensity (n = 102) | Rare | 17 | 16.67 |
| | Moderate | 33 | 32.35 |
| | Severe | 52 | 50.98 |
| Nuisance by time of day (n = 104) | Night | 84 | 80.77 |
| | Other | 20 | 17.09 |
| Location in house with highest nuisance (n = 104) | On bed & / or in bedroom | 93 | 79.49 |
| | Kitchen & other | 11 | 9.40 |
| Season with highest nuisance (n = 104) | Wet season | 76 | 73.08 |
| | Dry season | 17 | 16.35 |
| | Other | 11 | 10.58 |
| Insecticide used in household (n = 104) | Yes | 84 | 80.77 |
| | No | 20 | 19.23 |
| Insecticide form (n = 81) | Liquid | 64 | 79.01 |
| | Powder | 7 | 8.64 |
| | Other | 10 | 12.35 |
| Commercial name of insecticide identified (n = 84) | Yes | 47 | 52.22 |
| | No | 37 | 41.11 |
| Insecticide used within last two months (n = 104) | Yes | 48 | 40.68 |
| | No | 38 | 32.20 |
| | Do not know | 18 | 15.25 |
| Flea control (n = 104) | Yes | 80 | 67.80 |
| | No | 24 | 27.12 |
| Cockroach control (n = 104) | Yes | 33 | 27.97 |
| | No | 71 | 60.17 |
| Mosquito control (n = 104) | Yes | 9 | 7.63 |
| | No | 95 | 80.51 |

infrequently observed. House floor cleaning was done daily for 83.65% of respondents. Walls were generally constructed of sun-dried or baked clay bricks, or of mud blocks. In some homes, interior and exterior walls were plastered with a mixture of sand, mud, and/or cement. Roofs were either baked clay tiles (34.62%), thatched (33.65%), or corrugated iron sheet (31.73%).

Rodents were reported in 75.96% of homes, dominated by the house mouse (*Mus musculus*) (69.23%) and the black rat (*Rattus rattus*) (25.96%). Flea nuisance was a common problem in the communities (Table 2), with 50.98% of heads of households reporting that they or family members experienced severe flea nuisance (bites, scratches or the sensation of fleas crawling on body) in the last two months. Flea nuisance was mostly experienced at night (80.77%) and in bed and/or in the bedroom (79.49%). In addition, 73.03% of heads of households reported experiencing more intense flea nuisance with warmer temperatures (wet season). Domestic insecticide use was a common practice (Table 2), with 58.81% of households having used insecticide in the last two months and 80.77% having used chemical insecticide to control household pests at some point. Most participants bought insecticide from the local market and only 52.22% could give the name (brand or commercial name) of insecticide used, 79.01% of which were bought in liquid form and 8.64% as powder. The primary target of domestic insecticide treatments were fleas (67.80%), followed by cockroaches (27.97%), and mosquitoes (7.63%).

Candle traps were set in rooms according to head of household directive, with 65.38% placed on the 1st floor, 24.04% in the attic, and 10.58% on the ground floor. The head of

**Table 3. Distribution of flea species per village and per season in plague-endemic areas of Madagascar.**

| Village | Dry season | | | | | Wet season | | | | |
|---|---|---|---|---|---|---|---|---|---|---|
| | *P.i.* | *C.f.* | *E.g.* | *T.p.* | Total | *P.i.* | *C.f.* | *E.g.* | *T.p.* | Total |
| Alakamisy- Ambohimaha | 745 | 5 | 0 | 0 | 750 | 852 | 0 | 0 | 1 | 853 |
| Ambohipanalinana | 997 | 8 | 1 | 0 | 1006 | 1266 | 19 | 0 | 0 | 1285 |
| Nanda | 1783 | 5 | 4 | 0 | 1792 | 1905 | 95 | 3 | 0 | 2003 |
| Soafandry | 904 | 12 | 2 | 0 | 918 | 730 | 10 | 1 | 4 | 745 |
| Total | 4429 | 30 | 7 | 0 | 4466 | 4753 | 124 | 4 | 5 | 4886 |

*P.i.: Pulex irritans*, *C.f.: Ctenocephalides felis*, *E.g.: Echidnophaga gallinacea*, *T.p.: Tunga penetrans*

household usually chooses the bedroom (74.36%). In some instances, the room in which the trap was set served as a bedroom and kitchen (20.51%), as a spare room where nobody was sleeping (4.27%) or kitchen (1.70%).

A total of 9,352 fleas were collected from 126 houses investigated, with 98.18% (n = 9,182) being *P. irritans* and the remainder identified as *C. felis* (n = 154), *E. gallinacea* (n = 11), and *T. penetrans* (n = 5). Flea species distribution per village and per season is summarized in Table 3. Household flea prevalence was 99.03% during the dry season and 98.08% during the rainy season. The number of fleas collected did not differ per village when compared between seasons (Fig 2).

Since *P. irritans* represented >98% of fleas recovered, only number of *P. irritans* was considered as outcome variable in both models. The mixed model of flea abundance (*P. irritans* count; Model 1, Table 4) demonstrated a strong association with household size, where

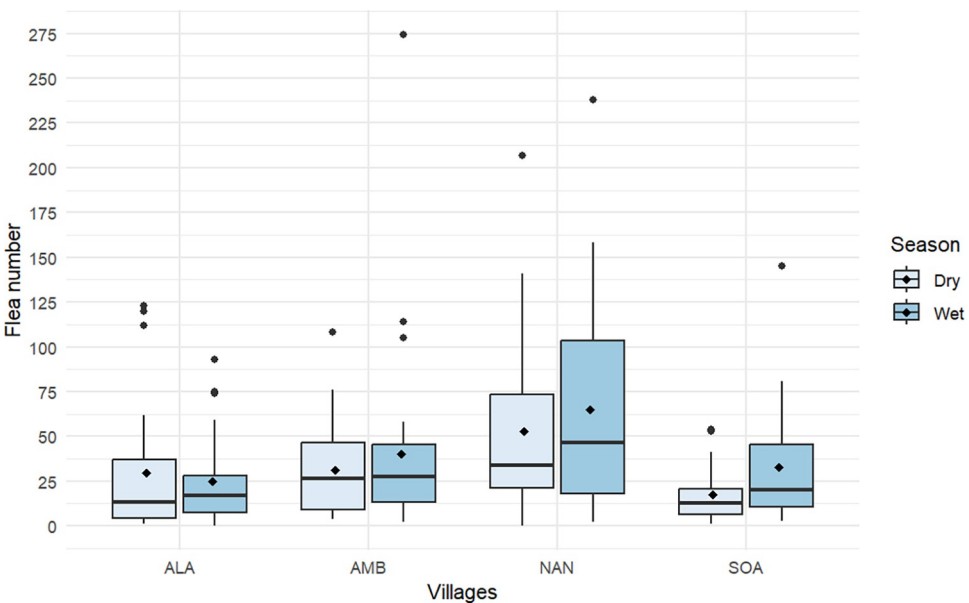

**Fig 2. Boxplot comparing flea number per village between seasons in plague-endemic areas of Madagascar.** ALA: Alakamisy Ambohimaha (p-value = 0.461), AMB: Ambohipanalinana (p-value = 0.435), NAN: Nanda (p-value = 0.530), SOA: Soafandry (p-value = 0.904). Black diamond-shaped points inside the boxes are mean values. Horizontal bars in boxes are the 50th percentiles (medians), and the bottom and the top of the box represent the 25th and the 75th percentiles, respectively. The two limits of vertical lines above and at the bottom of the box are the whiskers and represent the maximum and the minimum values of the data. Points outside the limit of vertical line are "outlier," which are values outside 95% of the confidence interval.

**Table 4. Factors associated with flea infestation in households in plague-endemic areas of Madagascar.**

| Variables | Levels | Flea abundance | | | | Infestation at high risk for interhuman *Y. pestis* transmission | | | |
|---|---|---|---|---|---|---|---|---|---|
| | | aOR | p-value | Low CI | High CI | aOR | p-value | Low CI | High CI |
| Household size | ≤4 (ref) | – | – | – | – | – | – | – | – |
| | >4 | **1.92** | **0.03** | **1.08** | **3.41** | – | – | – | – |
| Roof type | Elaborate* (ref) | – | – | – | – | – | – | – | – |
| | Primitive | 1.38 | 0.30 | 0.75 | 2.54 | 3.86 | 0.18 | 0.59 | 34.78 |
| Floor type | Mat¥ (ref) | – | – | – | – | – | – | – | – |
| | Other | 0.72 | 0.32 | 0.37 | 1.37 | **0.09** | **0.04** | **0.01** | **0.70** |
| Head of household finished primary school | Yes (ref) | – | – | – | – | – | – | – | – |
| | No | 0.91 | 0.74 | 0.53 | 1.56 | **0.19** | **0.05** | **0.03** | **0.92** |
| Any household member sleeping on floor | No (ref) | – | – | – | – | – | – | – | – |
| | Yes | 1.10 | 0.71 | 0.67 | 1.81 | 1.59 | 0.55 | 0.34 | 7.64 |
| Any household member sleeping under ITN | Yes (ref) | – | – | – | – | – | – | – | – |
| | No | 0.94 | 0.82 | 0.55 | 1.61 | 0.35 | 0.25 | 0.05 | 1.94 |
| Ever used insecticide | Yes (ref) | – | – | – | – | – | – | – | – |
| | No | 1.23 | 0.42 | 0.74 | 2.05 | **7.39** | **0.02** | **1.53** | **47.33** |
| Reported rodent presence | No (ref) | – | – | – | – | – | – | – | – |
| | yes | 1.66 | 0.07 | 0.96 | 2.87 | 6.64 | 0.07 | 1.06 | 70.64 |
| Chickens indoors at night | No (ref) | – | – | – | – | – | – | – | – |
| | Yes | **1.75** | **0.03** | **1.06** | **2.88** | 4.20 | 0.08 | 0.91 | 24.01 |
| Cow indoors at night | No (ref) | – | – | – | – | – | – | – | – |
| | Yes | 1.71 | 0.08 | 0.94 | 3.13 | 0.77 | 0.78 | 0.11 | 4.81 |
| Pigs indoors at night | No (ref) | – | – | – | – | – | – | – | – |
| | Yes | 0.99 | 0.98 | 0.50 | 1.95 | 4.71 | 0.16 | 0.59 | 45.61 |
| Village | ALA (ref) | – | – | – | – | – | – | – | – |
| | AMB | – | – | – | – | 0.21 | 0.29 | 0.01 | 3.43 |
| | NAN | – | – | – | – | 0.51 | 0.57 | 0.05 | 5.22 |
| | SOA | – | – | – | – | 0.06 | 0.06 | 0.00 | 0.86 |

*Elaborate roofs are made of corrugated iron sheets or clay tiles. Primitive roofs are made of thatch. Observation numbers for each category are in table 1.

¥ Floors covered with mats are usually earthen floor type. Other floor types are concrete, concrete mixed with broken tiles and wooden board. Ref: reference factor.

Bolded terms are significant at the p ≤ 0.05 level. aOR: adjusted odds ratio, CI: confidence interval.

households with more than four members had 1.92 times increased flea counts compared to households with fewer than four members, ($p = 0.03$, 95%CI: 1.08–3.41; Table 4). Households keeping chickens indoors at night had 1.75 times higher *P. irritans* count compared to those not keeping chickens indoors ($p = 0.03$, 95% CI: 1.07–2.28; Table 4). Households with rodent activity and keeping cows indoors at night (Table 4) also had increased flea counts though these estimates were marginally non-significant, indicating that increased precision through larger sample sizes may improve our estimate.

To emphasize the potential epidemiologic significance of observed *P. irritans* infestation, the model dichotomizing households into high and low interhuman *Y. pestis* transmission risk utilized a threshold for $P_{ii} = 7.43$ (CI 95%: 7.31–7.55; Table 4) as simulated by vector competence modeling. Thirty-four out of 104 (32.70%) households exhibited a $P_{ii}$ over 7.43, suggesting increased risk for sustained interhuman *Y. pestis* transmission based on vector competence modeling. Households which never used insecticides for pest control had increased odds of higher infestation risk compared to those which had used insecticides (aOR = 7.39, $p = 0.02$,

95% CI: 1.53 -– 47.33; Table 4). Households with floors made of concrete or board, as opposed to traditional fiber mats had lower odds of being in high-risk infestation households (aOR = 0.085, *p* = 0.04, 95% CI: 0.01–0.70; Table 4). Surprisingly, households with heads who had not finished primary school had lower odds of high infestation risk (aOR = 0.19, p = 0.05, 95% CI: 0.03–0.92; Table 4). A large aOR was found for primitive roof types (*i.e.*, thatch), indicating increased odds higher *Y. pestis* transmission risk, though the effect was non-significant with wide confidence intervals (aOR = 3.86, p = 0.18, 95% CI: 0.59–34.78; Table 4). Households keeping chickens indoors at night and reporting rodent activity also had increased odds of high infestation risk, though these estimates were marginally non-significant (aOR = 4.20, p = 0.08, 95% CI: 0.01–24.0 for indoor chickens; aOR = 6.64, p = 0.02, 95% CI: 1.53–47.33 for rodent activity; Table 4), suggesting that improved precision is necessary to better estimate this association. These results were robust to the *m* simulation for *Pii* thresholds (S4 File).

## Discussion

Our observations of high household flea infestation rates dominated by the human flea *P. irritans* reinforce the findings of previous studies in rural Madagascar [19,22,23,36]. Furthermore, flea number captured per household in our study were high compared to adjusted averages reported in other countries using comparable sampling techniques [18,37–39]. Consistent with other studies, the number of fleas was found to be highly variable among households within the same village emphasizing the likely role of household characteristics [19,23].

Madagascar is one of the countries where plague remains a public health concern. During the last two decades, there have been more than 13,000 human plague cases in Madagascar with a fatality rate of ~27% [27]. Our findings suggest that households in plague-endemic areas of Madagascar are frequently but heterogeneously infested by *P. irritans*, a flea species that has been found infected by the plague bacterium during previous epidemics in Madagascar and elsewhere [20,22,40]. Although the vector capacity of *P. irritans* is unknown, in Tanzania, the density and distribution of *P. irritans* was associated with plague frequency and plague incidence [18]. Laboratory transmission tests showed that *P. irritans* was a less potent vector than rodent fleas [20,21] but high household flea burden may facilitate an interhuman transmission event [20].

Based on laboratory data [21] and flea transmission modeling using the vectorial capacity equation for early-phase *Y. pestis* transmission following published procedures [17], a household infestation of more than seven *P. irritans* per person per household ($P_{ii}$>7.43) was estimated to potentially sustain interhuman transmission of *Y. pestis* should the pathogen be present in this vector-host ecosystem. A third of surveyed households were in the higher risk category of high $P_{ii}$, presenting a considerable threat of sustained *Y. pestis* transmission vectored solely by *P. irritans*. It is unlikely that a plague outbreak would be sustained solely by *P. irritans*, however, while the proportion of transmission events attributed to *P. irritans* may be low in the presence of more competent vectors, this additional route deserves more attention from epidemiologists. Since *P. irritans* was the most abundant species found in human domiciles, modeling based on this assumption was a first step toward exploring the role which *P. irritans* may play in local outbreaks. [20]

This study identifies factors associated with increased flea abundance that also place households in this plague-endemic area at increased risk of sustained transmission if flea were exposed to circulating *Y. pestis*. The results suggest several modifiable environmental features, including household construction materials associated with high flea infestation.

Certain floor types (i.e., concrete and board) were negatively associated with flea abundance, providing a protective effect. Previous studies reported that thorough and frequent

floor cleaning could remove fleas of all developmental stages, as well as organic particles on which larvae feed, and thus could reduce flea infestations [4,41]. Most households reported daily floor sweeping (Table 1), although the dirt floor under the mat covering would be left undisturbed. The use of smoother flooring materials, such as board or concrete, may allow more thorough cleaning and may explain the protective effect demonstrated in our model. Indeed, studies involving other flea species mentioned earthen floor as one of the risk factors for the prevalence of flea infestations or flea-induced skin disorders [25,42–44]. Flea immature stages are very susceptible to desiccation [2,45], and earthen floor covered with plant fiber mats may offer immature fleas the best conditions for survival.

Although the results did not achieve statistical significance and lack precision, the substantial effect size observed for primitive roof types (e.g., thatch) being linked to an increased likelihood of falling into the higher *Y. pestis* transmission risk category implies a potentially noteworthy role for this construction material. However, it is essential to note the need for further investigation to validate these preliminary findings. Organic material such as thatch may offer a more stable environment for insect development, as reported for other vectors [46]. Interestingly, another study that examined plague epidemiological data and household characteristics also associated thatched roof with human plague risk [26]. Although flea abundance was not among factors studied, thatched roofs favored human contact with the black rat and their fleas.

House construction materials may also reflect household income. In this study, individual household gross income was not assessed directly, but the national census reported that a mat floor and thatch roof are among the housing material characteristics of the poorest households in Madagascar [47]. With 88% of the rural population living under the International Poverty Line, house building materials choice would be biased toward locally sourced and thus, more affordable materials. Therefore, any action to alleviate flea burden should prioritize the most vulnerable households in the community. On a national scale, 77.8% of the Malagasy rural population live in houses constructed of non-durable materials, including 41.8% woven plant fiber mats, and 66.5% plant-derived roof materials [47]. Since insecticide treatment has little effect on *P irritans* infestation [23], improved house construction programs offer the most promise for mitigating infestations. This study identified strategic home modifications against *P. irritans* infestation that also align with non-profit and government goals. Increasing the number of households with more durable and easier to clean floor types would likely benefit the entire community by reducing flea infestation prevalence at village level. As reported for programs targeting other vectors, house improvement generated other health benefits and increased inhabitant life quality [48].

Keeping any livestock indoors at night was among the risk factors identified. Although the odds ratios of keeping pigs and chicken indoors at night were non-significant with plague risk analysis (Model 2), the analysis of factors affecting flea abundance (Model 1) showed that keeping chickens indoors at night increases the odds of having *P. irritans* infestation (Table 4). Raising chicken was among the risk factor for house flea infestation in a study conducted in China [25] and chicken DNA was among the host genetic material detected in wild-caught *P. irritans* in DR Congo [49]. In this study, 66.35% of the households raised chicken and more than 60% kept them indoors at night. Therefore, keeping those animals in separate structures may alleviate flea burden. In Madagascar livestock housing choice may vary according to region, climate, and farming practices [50]. Unfortunately, in the study area, livestock were usually kept on the ground floor at night, due to concern of theft, which may promote flea infestation and increase the disease transmission risk. These animals are among potential hosts for adult *P. irritans* in Madagascar since this species has been collected in pig pens [30] and on chickens [51]. In other countries, this flea species has also been found infesting various

livestock including chickens and pigs, [3,5,52]. However, without host blood source identification from field-caught fleas, it is difficult to establish a clear link between flea abundance and any animal presence. In addition, animal waste was pointed out as a potential source of flea reinfestation in cattle since manure accumulation is a source of heat, humidity, and organic material favorable for flea larvae development [3,52].

Domestic insecticide use was highlighted as a common practice for flea control in our study. The model suggested that insecticide use against any household pest might be a factor that influenced flea infestation. The elevated prevalence of *P. irritans* infestation may elucidate the necessity for employing domestic insecticides to mitigate the perceived nuisance as reflected in Table 2. Nonetheless, our analysis revealed an elevated adjusted odds ratio with wide confidence interval (Table 4), emphasizing the need for caution in interpreting the possible effect due to the considerable uncertainty. Although our model showed that chemical insecticides may have a protective function against high flea abundance, there are concerns for the long-term efficacy of this method due to insecticide resistance. Previous studies in Madagascar suggested that insecticide treatment deployed during plague outbreaks were inefficient for the *P. irritans* [23,24]. However, insecticide resistance in *P. irritans* has never been investigated in Madagascar. This is especially concerning as most of study participants could not recall the name of insecticides used. This lack of household knowledge surrounding insecticide use could lead to mismanagement of chemicals that may induce insecticide resistance in *P. irritans* and other flea species over time.

Seasonal abundance among rodent flea has been correlated with climatic factors in the Central Highlands of Madagascar [45,53], with higher flea indices observed in the beginning of the rainy season, which coincides with the onset of bubonic plague transmission [54]. In this study a seasonal pattern for *P. irritans* abundance was not established despite interview respondents reporting more intense flea nuisance during the rainy season (Table 2). Since rodent flea abundance depends also on rodent host physiology and reproduction [36], *P. irritans* appear to benefit from a more stable home environment. However, our findings are limited by small sample size and the fact that our study represents only a snapshot in time for both seasons.

Rodent presence was examined in this study, since rodents are the host of many flea species, including *P. irritans* [55]. Interestingly, *P. irritans* has rarely been found infesting rodents in Madagascar and thus, might not be the primary host for this flea species [22,53]. Analysis of homeowner responses concerning perceptions of flea nuisance (Table 2) suggests a potential scenario where this species is feeding on humans. Interestingly, we found a positive association between household size and *P. irritans* abundance. Households with more than four members have, on average, 1.92 times higher odds of experiencing *P. irritans* infestation (Table 4). Consequently, the larger family size may offer an increased opportunity for fleas to access blood meals and sustain a larger flea population. Moreover, in larger households, diverse activities may increase flea exposure. Similar trends were observed in Ethiopia regarding tungiasis [43,44], and in Bogota regarding flea-induced skin disorder [42], indicating higher risks for children going to school, from larger families, and those using public transportation. Research in China suggests that floor fleas can transfer between houses, especially in larger villages [25]. Sampling fleas in shared spaces like schools, churches, and public transportation could be valuable for future investigations.

Our results demonstrated that the odds of a high flea infestation index varied by village. This effect could be due to the proportion of households per village presenting one or several risk factors. Although no village level factors were included in the present model, it has been demonstrated elsewhere that village size, distance between homes, proportion of households raising chickens, and presence of a central waste disposal area can influence the prevalence of

off-host fleas [25]. These village-level factors may influence *P. irritans* ecology and deserve to be investigated further in the future.

One of the main plague risk indicators is the flea index, which is obtained by dividing number of fleas by number of hosts sampled [56]. The same method was applied to obtain a house or nest index for off-host fleas [16]. The *P. irritans* index calculated in this study estimated human exposure to flea bites capable of sustaining plague transmission for each household. This index was obtained by dividing the total number of *P. irritans* collected during three successive nights by household size. A limitation of our study was that the sampling method underestimates the number of fleas collected in the household, since only a single room per household was sampled. Furthermore, our model assumed that members of the same household were equally exposed to flea bite risk, whereas the odds of being bitten by fleas may vary even between individuals within a household [5]. More extensive sampling, including more household rooms, would give a better estimate of a household's flea population. Another limitation of our study was the scarcity of *P. irritans* vector competence studies [21]. Our model was based on poor vector competence of *P. irritans* collected from owls and foxes, which may under-estimate vector competence of human-adapted strains that may bite more frequently [21]. Therefore, the strength of the model could be improved by incorporating more *in vivo* values of vector competence on the human-adapted strains from plague-endemic areas of Madagascar. Entomological parameters such as biting rates, host preferences, and daily survivorship of infected *P. irritans*, must also be further explored to quantify the role of *P. irritans* in plague epidemics in Madagascar.

## Conclusions

The present study confirms that *P. irritans* infestation is a neglected nuisance in rural households in Madagascar. Since this flea species does become naturally infected with the plague bacterium, further studies concerning its biology, ecology, and vector competence are wanted. Our results demonstrate that one third of investigated households in plague endemic areas of Madagascar were exposed to a high *P. irritans* index, putting them at risk of sustained interhuman plague transmission, should the fleas or a household member become infected. Furthermore, infestation risk was more pronounced for poor households living in homes made with organic materials, and in close contact with livestock. The factors identified in this analysis suggest that improvement of housing and better management of livestock would alleviate flea burden and potential plague risk in rural plague-endemic villages experiencing high flea infestation.

## Supporting information

**S1 File. Survey instrument in English.**
(PDF)

**S2 File. Photos of the traditional three-story house in the central highland of Madagascar with various roof type.**
(PDF)

**S3 File. Diagram of three-story traditional house in the central highland of Madagascar, with common use of each level.**
(PDF)

**S4 File. Sensitivity analysis of the infestation at high risk for interhuman *Y. pestis* transmission using *m* upper and lower cut-off values.**
(PDF)

## Acknowledgments

We thank Centre Valbio Ranomafana for logistical support; Andrianirina O. Rafanambinant-soa, Paul JN. Niaina, Jean-Francois A. Randrianasolo, Farida Juliette and Mandimby A, Rajao-narimanana, for assistance with flea collection and interview. We are also grateful to Belen Santana-Godinez for her contribution to the early conception of the project, to Pr. Josef Zeyer for his critical review and insightful comments which improved the manuscript, and to Dr. Mireille Harimalala for hosting the students working on this project in her research group at Institut Pasteur de Madagascar. We also want to extend our gratitude to the study participants and the authorities in the villages visited.

## Author Contributions

**Conceptualization:** Adélaïde Miarinjara, Stephen Gilbert Mugel, Nick An, Minoarisoa Esther Rajerison, Rindra Vatosoa Randremanana, Romain Girod, Thomas Robert Gillespie.

**Data curation:** Adélaïde Miarinjara, Annick Onimalala Raveloson, Stephen Gilbert Mugel.

**Formal analysis:** Adélaïde Miarinjara, Annick Onimalala Raveloson, Stephen Gilbert Mugel, Thomas Robert Gillespie.

**Funding acquisition:** Adélaïde Miarinjara.

**Investigation:** Adélaïde Miarinjara, Annick Onimalala Raveloson, Andry Andriamiadanarivo, Thomas Robert Gillespie.

**Methodology:** Adélaïde Miarinjara, Stephen Gilbert Mugel, Minoarisoa Esther Rajerison, Rindra Vatosoa Randremanana, Romain Girod, Thomas Robert Gillespie.

**Project administration:** Adélaïde Miarinjara, Thomas Robert Gillespie.

**Supervision:** Minoarisoa Esther Rajerison, Rindra Vatosoa Randremanana, Romain Girod, Thomas Robert Gillespie.

**Validation:** Adélaïde Miarinjara, Annick Onimalala Raveloson, Stephen Gilbert Mugel, Minoarisoa Esther Rajerison, Rindra Vatosoa Randremanana, Romain Girod, Thomas Robert Gillespie.

**Visualization:** Adélaïde Miarinjara, Stephen Gilbert Mugel, Thomas Robert Gillespie.

**Writing – original draft:** Adélaïde Miarinjara, Annick Onimalala Raveloson, Stephen Gilbert Mugel, Nick An, Andry Andriamiadanarivo, Minoarisoa Esther Rajerison, Rindra Vatosoa Randremanana, Romain Girod, Thomas Robert Gillespie.

**Writing – review & editing:** Adélaïde Miarinjara, Stephen Gilbert Mugel, Nick An, Andry Andriamiadanarivo, Minoarisoa Esther Rajerison, Rindra Vatosoa Randremanana, Romain Girod, Thomas Robert Gillespie.

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
