## [Decision Letter · Decision Letter 0]

6 Sep 2023

Dear Dr Miarinjara,

Thank you very much for submitting your manuscript "Socio-ecological risk factors associated with rural household human flea infestations in plague-endemic areas of Madagascar." for consideration at PLOS Neglected Tropical Diseases. As with all papers reviewed by the journal, your manuscript was reviewed by members of the editorial board and by several independent reviewers. In light of the reviews (below this email), we would like to invite the resubmission of a significantly-revised version that takes into account the reviewers' comments. 

We cannot make any decision about publication until we have seen the revised manuscript and your response to the reviewers' comments. Your revised manuscript is also likely to be sent to reviewers for further evaluation.

Sincerely,

Benjamin L. Makepeace

Academic Editor

Victoria Brookes

Section Editor

Please review the referees' comments carefully, especially the critique of reviewer 2 regarding the choice of statistical method, which does seem inappropriate. The revised manuscript must include reanalysis with alternative approaches.

Reviewer's Responses to Questions

**Key Review Criteria Required for Acceptance?**

**Methods**

-Are the objectives of the study clearly articulated with a clear testable hypothesis stated?

-Is the study design appropriate to address the stated objectives?

-Is the population clearly described and appropriate for the hypothesis being tested?

-Is the sample size sufficient to ensure adequate power to address the hypothesis being tested?

-Were correct statistical analysis used to support conclusions?

-Are there concerns about ethical or regulatory requirements being met?

Reviewer #1: Nice use of candle trap! 

Is the R code available with the data? I would strongly recommend that it is, either as part of suppl data or at any of the servers with doi or github.

Reviewer #2: Major critique:

Stepwise regression should almost never be used. The selection of variables to include in a regression model based on their individual p-values is normally bad practice, even though it is a practice that has been proposed and is still used (perhaps especially in some biomedical journals). There is now quite a bit of literature about why the approach is normally inappropriate, why it leads to analytic and statistical problems, and the different types of problems that are inherent in this approach. 

While this article is in the "Journal of Big Data", the logic is relevant for smaller datasets as well and the bibliography lists many of the papers that discuss problems with the approach. 

https://journalofbigdata.springeropen.com/articles/10.1186/s40537-018-0143-6

Here are a few other accessible articles that likewise include references to the relevant literature:

https://towardsdatascience.com/stopping-stepwise-why-stepwise-selection-is-bad-and-what-you-should-use-instead-90818b3f52df

https://statmodeling.stat.columbia.edu/2014/06/02/hate-stepwise-regression/

Briefly, a better approach to regression would begin with scientific hypotheses about which variables are likely important for the process that you are trying to model. Presumably, many of the variables from the survey were included in the survey because the researchers thought they could be important.

One of the larger problems with stepwise regression then comes from its deviance from the scientific approach. If you have a variable that you think should be important for the process you're modeling, and it is not showing a statistical association - that finding is of scientific value. It is equally as important (maybe even moreso) than a statistically significant finding. A related issue is that sometimes variables in a regression modify the association between other variables and the outcome of interest - EVEN WHEN they are not statistically significant. Conversely, another issue is that some variables that are not statistically significant on their own can become statistically significant in a regression model, after other factors/variables have been accounted for. 

I suggest starting with the variables that you think should be important (hopefully based on scientific hypotheses or logic). Generate a table with their odds ratios and confidence intervals (or other relevant statistics - such as abundance/counts/risk ratios, etc.) Then include the model adjusted odds ratios and confidence intervals in the table as well. IF you have variables that are redundent, then it could be good to exclude all but one from the model based on that redundency (not based on p-values). You can often tell which ones are redundent by looking at how they co-vary (are they essentially always the same in the same households?), through logical reasoning, or by model behavior.

Another issue that I see has to do with the model that was used (binarly generalized linear model, which most folks would call a logistic regression). The approach begins with Equation 1 - which is an equation for the number of fleas per person required to sustain transmission, under a host of assumptions (including a major one of R0 = 1). I believe the authors need to either re-think this step or provide further justification for its use. It depends heavily on quite a lot of assumptions, and furthermore i don't think all of the assumptions or this approach are necessary. We already know that there is ongoing disease risk in this setting. Why rely on a theoretical model that would normally only be applied to larger populations? For example, R0 is normally only useful as a concept for larger, homogeneously mixing populations. From this equation, a continuous variable is generated. This continuous variable is then categorized as binary (0 or 1) based on a hypothesized threshold over which there is greater risk of infection. I see another problem with this step, in that risk is not binary and the threshold for high versus low risk appears relatively arbitrary. 

My proposal then would be to model the abundance of fleas. This could also be done using a generalized additive model, but would instead be a type of count model (Poisson, negative binomial, etc., depending on the data). It would be possible and important to account for household occupancy still. The regression could account for repeat observations within households across nights and seasons using a random intercept at the household level (i.e. each household has its own intercept). Seasonal variation could then be measured in abundance, while accounting for the other factors that might be of importance. I believe that this would be a better analytic approach, and that it would still achieve the goal of this research: to indentify factors that contribute to risk of infection by rodent/flea-borne disease - with an inherent assuption that more fleas per person is tantamount to more risk of flea-borne disease per person. 

Another approach could be to model 'm' from Equation 1 as a Gaussian GLM, but the approach that I suggest above (Poisson or negative binomial, with counts of fleas per house and a random intercept per house) would likely be more straightforward and would rely on many fewer assumptions. 

Minor issues:

Line 73: "...plague is the most infamous." I don't doubt this, but since this is a scientific manuscript it is probably better to only include statements that can be supported, or to soften the statement. One solution could be to include: "...plague is arguably the most infamous" Or otherwise the authors could include a reference that shows that plague is in fact the most infamous (surveys?)

Line 79: State what is mean by 'humans are most vulnerable' Does it mean they are more likely to come into contact with rodents or fleas? To be infected? To have severe disease?

Line 103: Household level analyses are valuable and there could be household level interventions - but the results from household level analyses could also lead to community level interventions, whereby community engagement approaches community members with information about household level correlates of fleas.

Line 119: Does the fact that interviews were done in the dry season influence the results? Might respondents have answered differently if they'd been interviewed in the wet season? Is this also why there were very few male respondents?

Line 130: How were the surveys developed? Were they based on pre-existing surveys?

Line 138: How were households selected for inclusion in this study?

Line 249: The authors probably mean 'multivariable analysis" instead of 'multivariate analysis'. A multivariate regression would mean that there were different outcomes being modeled. I believe here there is a single outcome being modeled, with muliple predictor variables in the model. This would be a multivariable regression.

Line 255: I'm not sure what is meant by "our analysis did not demonstrate statistical significance." It looks as though several variables are statistically significant. 

Along the same lines, it would be better to present model-adjusted odds ratios (or rate ratios with a Poisson or negative binomial regression) and their confidence intervals. No p-values are needed if we have the effect size and confidence intervals, and these statistics are much more informative than the p-values. It is good to not rely too heavily on p-values as few people really understand them.

Reviewer #3: Are the objectives of the study clearly articulated with a clear testable hypothesis stated? - yes

-Is the study design appropriate to address the stated objectives? - yes

-Is the population clearly described and appropriate for the hypothesis being tested? - yes

-Is the sample size sufficient to ensure adequate power to address the hypothesis being tested? - yes

-Were correct statistical analysis used to support conclusions? - yes

-Are there concerns about ethical or regulatory requirements being met? - no

**Results**

-Does the analysis presented match the analysis plan?

-Are the results clearly and completely presented?

-Are the figures (Tables, Images) of sufficient quality for clarity?

Reviewer #1: Was the night irritation/bits truly linked to fleas. Were bedbugs considered? Often bedbug are not known and blame goes onto fleas. Not self-reporting bedbugs does not mean they not there. 

I did not find the information about the animals within household, descriptive data. The floor mats and dirt makes perfect sense with pigs indoors. This is perfect for Pulex emergance. The lesser # of Ctenocephalides is likely that dogs would not be allowed inhouse unlike pigs that would be spending more time there, so depositing flea eggs on he floor mats to develop. You OR for pigs indoors includes 1 and that is really unexpected for me, could you granulate the "livestock" what other livestock is there? What about daytime? Where are the pigs traditionally? Inside or outside during the day? Or are they allowed inside during the day?

Reviewer #2: Please see my comments in the methods section, which will have relevance to this section

Reviewer #3: Does the analysis presented match the analysis plan? yes

-Are the results clearly and completely presented? yes

-Are the figures (Tables, Images) of sufficient quality for clarity? yes, but table 4 is in the wrong place

**Conclusions**

-Are the conclusions supported by the data presented?

-Are the limitations of analysis clearly described?

-Do the authors discuss how these data can be helpful to advance our understanding of the topic under study?

-Is public health relevance addressed?

Reviewer #1: The recommendations are sound and completely fair given the data authors had. Authors should point that the fleas that emerge are only those that developed in the household, fleas don't jump off the animal. So what you detected is what developed within the trap. Placing the trap in areas where animals spent most of their time is beneficial, because they will usually not emerge (triggered by the light/heat) further then say 2 feet away. Pulex will be primary from pigs in this context, hence my question before about the pigs inside and time they spent there day/night. While you cannot go back you can provide cultural context beyond just saying they keep them in at night due to theft. I would be expecting that pigs might be simply allowed to be inside the house all the time and in fact rest there as well.

Reviewer #2: Please see my comments in the methods section, which will have relevance to this section

Reviewer #3: -Are the conclusions supported by the data presented? yes

-Are the limitations of analysis clearly described? yes, but can be enriched

-Do the authors discuss how these data can be helpful to advance our understanding of the topic under study? yes

-Is public health relevance addressed? yes but can be expanded

**Editorial and Data Presentation Modifications?**

Reviewer #1: Make the data and code available via doi.

Reviewer #2: Please see my comments in the methods section, which will have relevance to this section

Reviewer #3: Review of the manuscript entitled “Socio-ecological risk factors associated with rural household human flea infestations in plague-endemic areas of Madagascar.”

The manuscript is presenting a study with two main components. First, analysis of data from field data collections in Madagascar of human fleas in rural households across dry and wet seasons in 4 villages is done. This elucidates the association of possible risk factors with high flea infestation. Second, the researchers propose a model to estimate the minimum number of fleas per person necessary to sustain plague transmission once the pathogen enters the human-flea ecosystem.

Overall comment: this research adds important insights to the field and adds valuable information to the mounting evidence that the human flea could also be involved in plague outbreaks even if only in a minor role. The description of the environment Malagasy live in, in rural areas, and how they are affected by and deal with human flea infestations is very informative. I would suggest elaborating more on this to enable readers to understand the conditions better. Explaining the living conditions and what kind of livestock is usually kept where is important (such as broody hens in the bedroom). 

Major comments:

In the background section:

Traditional housing is associated with flea infestation, so maybe add a short paragraph here on living conditions and traditional housing in rural Mada. % of people living with reed roofs, mats, livestock below (which livestock species are in the houses), fleas are abundant, sometimes rooms/floors are abandoned due to flea infestation. 

Table 4 belongs into the results section.

Please discuss in more detail the use of the mats on the ground, namely as an affordable substitute for furniture (i.e. people sitting on the mats during meals, sleeping on mats, children playing on the mats). Following this it is important to also later discuss the difficulties of discouraging their use and suggest ways of improving the situation (exposing them to the sun, dusting with strong insecticide etc.). Also the possibility of government interventions such as IRS. 

Please add to your limitations that trapping in the same type of room (i.e. bedroom) across all households would standardize data more. Also, that trapping across 2 wet seasons but only 1 dry season may have affected the results on seasonality.

Apart from lab studies to learn more about P. irritants, I suggest you also recommend research on possible interventions. 

Minor comments:

Please consider changing the title to:

Socio-ecological risk factors associated with human flea infestations of rural household in

plague-endemic areas of Madagascar.

Page 2 line 32 change to “risk factors associated with such…”

Background:

Please add a sentence explaining the variable host specificity of fleas. (ref 10).

Page 5 line 90-92: what do want to state by saying other plague reporting countries do not report this magnitude of human fleas? Do you mean that is why Mada may have more and more prolonged outbreaks? Explain please.

Page 6 line 126 informed oral consent or written consent?

Page 8 line 164: explain threshold septicemia.

Page 8 line 168 you need to define household size to avoid misunderstanding. People per household versus floor area in m2

page 8 line 174: categorical variable levels like what? Can you give an example?

Table 1: did you ask how often insecticide was used?

How did you deal with the connection between owning livestock and keeping livestock?

Results:

Please explain clearer why keeping livestock would increase P. irritans density.

Page 10 line 192: women or men more likely to finish primary school?

Line 193 sleeping on the floor on what?

page 11 203 covered with a woven plant….

Line 208 consider calling it corrugated iron sheet instead of metal sheet.

Line 209 …mainly reported to be the house mouse….

Line 215 please stick to wet and dry or explain earlier that warm and wet and cool and dry goes together. Otherwise the reader cannot follow here. 

Page 12: any info on what active ingredient is in the commonly sold insecticides?

Page 14 line 256 ..was found to be lower…

Line 259 …found for households 

Discussion

Why is table 4 in the discussion? Should be in the results section

Page 17 line 307 delete Conversely

Consider mentioning a possible connection between the use of mats and thatched roofs, because they are both indicators of low socio-economic status.

Consider rewriting the livestock part in the discussion to clarify what species are usually kept per household (i.e. every hh has chickens), where they are kept etc.

Page 20 line 363 delete susceptible

Line 379 present tense: demonstrate

Conclusion

Page 21 line 389: …competence are needed…

Line 391 change to: …index. According to the model estimates this puts them….

**Summary and General Comments**

Reviewer #1: This is interesting study that I enjoyed reading. What surprised me was that Pulex was considered, in introduction you rightly discounted Ctenocephalides due to being poor vector of Yersinia. Is Pulex actually good vector? There some old works that compared vectorial capacity of different fleas, could you provide the insight and evidence how it compared to say Oriental rat flea? You elude to it in discussion, but some of it might be suited for introduction as well and I am surprised you say no one knows if Pulex can transmit Yersinia - I believe I read an old study about that, but I can be incorrect.

Reviewer #2: This manuscript presents results from an analysis of data on flea abundance, household characteristics, and and other household-level socio-economic factors thought to be associated with rodents, fleas, and rodent-borne disease. The topic and described data appear quite interesting and relevant. I've got several minor comments and critiques and one major one, with the latter focused on the statistical approach.

Reviewer #3: Overall comment: this research adds important insights to the field and adds valuable information to the mounting evidence that the human flea could also be involved in plague outbreaks even if only in a minor role. The description of the environment Malagasy live in, in rural areas, and how they are affected by and deal with human flea infestations is very informative. I would suggest elaborating more on this to enable readers to understand the conditions better. Explaining the living conditions and what kind of livestock is usually kept where is important (such as broody hens in the bedroom).

PLOS authors have the option to publish the peer review history of their article (what does this mean?). If published, this will include your full peer review and any attached files.

Reviewer #1: No

Reviewer #2: No

Reviewer #3: No
---

## [Decision Letter · Decision Letter 1]

13 Feb 2024

Dear Dr Miarinjara,

Thank you very much for submitting your manuscript "Socio-ecological risk factors associated with human flea infestations of rural household in plague-endemic areas of Madagascar." for consideration at PLOS Neglected Tropical Diseases. As with all papers reviewed by the journal, your manuscript was reviewed by members of the editorial board and by several independent reviewers. The reviewers appreciated the attention to an important topic. Based on the reviews, we are likely to accept this manuscript for publication, providing that you modify the manuscript according to the review recommendations. 

Sincerely,

Benjamin L. Makepeace

Academic Editor

Victoria Brookes

Section Editor

The reviewers and I find the manuscript to be greatly improved and appreciate the effort made to reanalyse the data. However, reviewer 2 highlights some important issues with interpretation that must be addressed by revisions to language in the paragraphs they critique below.

Reviewer's Responses to Questions

**Key Review Criteria Required for Acceptance?**

**Methods**

-Are the objectives of the study clearly articulated with a clear testable hypothesis stated?

-Is the study design appropriate to address the stated objectives?

-Is the population clearly described and appropriate for the hypothesis being tested?

-Is the sample size sufficient to ensure adequate power to address the hypothesis being tested?

-Were correct statistical analysis used to support conclusions?

-Are there concerns about ethical or regulatory requirements being met?

Reviewer #1: (No Response)

Reviewer #2: Yes

**Results**

-Does the analysis presented match the analysis plan?

-Are the results clearly and completely presented?

-Are the figures (Tables, Images) of sufficient quality for clarity?

Reviewer #1: (No Response)

Reviewer #2: Yes, but see my notes

**Conclusions**

-Are the conclusions supported by the data presented?

-Are the limitations of analysis clearly described?

-Do the authors discuss how these data can be helpful to advance our understanding of the topic under study?

-Is public health relevance addressed?

Reviewer #1: (No Response)

Reviewer #2: Yes, but see my notes

**Editorial and Data Presentation Modifications?**

Reviewer #1: (No Response)

Reviewer #2: (No Response)

**Summary and General Comments**

Reviewer #1: (No Response)

Reviewer #2: The authors have done a great job of addressing my previous critiques. I appreciate that they've added the new model I suggested (modeling counts of fleas), and am satisfied with their argument to keep the older model in as well.

Overall I think this leads to a much improved manuscript. I have a few remaining critiques, that are likely holdovers from the older manuscript and need to now be updated. 

Major:

A few of the variables (especially with regard to house materials) were not statistically significant in the models. In particular, this means that their confidence intervals crossed the 1.0 threshold, and that it isn't really possible to say whether those associations were negative or positive. This doesn't exclude the possibility of a true ecological association, and it is possible that the reason for not being statistically significant is because of a lack of power. There are other possibilities as well. 

Line 365 - please soften this statement since we can't be sure whether the association was negative or positive, given the confidence intervals. There is a lot of uncertainty in that point estimate. You could mention that the odds ratio is large and positive, but that the confidence intervals crossed the threshold so that no certain statements can be made about the association between this variable and the outcome.

Line 383: Do note that there was an effect of having chickens indoors and the abundance of fleas...

Line 398: We can't be sure from this analysis that a larger sample size would actually reveal the hypothesize association. If the authors wanted to, they could pursue a post-hoc power analysis and discuss those results along these lines. I would suggest just softening these statements about ORs that had incredibly broad CIs.

Also, I note from Table 4 that there is a positive association between reported insecticide use and the odds of having an 'infestation' of fleas in the house. I've seen similar results previously for other disease systems, where individuals who are experiencing lots of exposure to arthropods are most likely to use insecticides to address that exposure. The results from a cross sectional survey can then make the association appear causal in the opposite direction - where houses with more insects are the most likely to use insecticide (though the insecticide was unlikely to be the cause of the insect infestation). Regardless, I suggest rewording the statements from line 398 with this in mind, and also with the results from Table 4 in mind. 

Minor: 

Line 110 - there is a mention that in Madagascar, during plague outbreaks, there are human flea infestations 'at magnitudes rarely found in other...' - I suggest specifying what is meant here explicitly. Perhaps something along the lines of:"...frequently infested by large numbers of human fleas..." This might help this statement be a bit more clear.

Line 208: Please use "generalized" for the generalized linear model. 

Line 211 - there may be a typo in between "m. Seasonality"

Lastly, I found the positive association between numbers of humans in a house and numbers of fleas in a house to be interesting, especially given that the population dynamics of these anthropophilic fleas should be associated with the numbers of hosts to which they have access. Perhaps this association is obvious, but I would at least mention it in the discussion because it is a valuable finding and could potentially push the referenced modeling work further (i.e. there should/may be feedbacks between human and flea population dynamics).

PLOS authors have the option to publish the peer review history of their article (what does this mean?). If published, this will include your full peer review and any attached files.

Reviewer #1: No

Reviewer #2: No

Figure Files:

Data Requirements:

Reproducibility:

References

---

## [Editor Report · Decision Letter 2]

29 Feb 2024

Dear Dr Miarinjara,

We are pleased to inform you that your manuscript 'Socio-ecological risk factors associated with human flea infestations of rural household in plague-endemic areas of Madagascar.' has been provisionally accepted for publication in PLOS Neglected Tropical Diseases.

Best regards,

Benjamin L. Makepeace

Academic Editor

Victoria Brookes

Section Editor

---

## [Editor Report · Acceptance letter]

4 Mar 2024

Dear Dr Miarinjara,

We are delighted to inform you that your manuscript, "Socio-ecological risk factors associated with human flea infestations of rural household in plague-endemic areas of Madagascar.," has been formally accepted for publication in PLOS Neglected Tropical Diseases.

Best regards,

Shaden Kamhawi

co-Editor-in-Chief

Paul Brindley

co-Editor-in-Chief
